# Effect of Asymmetric Accumulative Roll-Bonding process on the Microstructure and Strength Evolution of the AA1050/AZ31/AA1050 Multilayered Composite Materials

**DOI:** 10.3390/ma13235401

**Published:** 2020-11-27

**Authors:** Sebastian Mroz, Arkadiusz Wierzba, Andrzej Stefanik, Piotr Szota

**Affiliations:** 1Faculty of Production Engineering and Materials Technology, Czestochowa University of Technology, Av. Armii Krajowej 19, 42-201 Czestochowa, Poland; andrzej.stefanik@pcz.pl (A.S.); piotr.szota@pcz.pl (P.S.); 2Metalurgia S.A., Świętej Rozalii 10/12, 97-500 Radomsko, Poland; wierzbaarkadiusz@gmail.com

**Keywords:** magnesium alloy, aluminum, multilayered materials, asymmetric accumulative roll-bonding (AARB), microstructure, strength, FEM analysis

## Abstract

This paper aimed to propose the fabrication of light, Al/Mg/Al multilayered composite. Initially prepared three-layered feedstock was subjected to deformation during four rolling cycles (passes) using the conventional and modified accumulative roll bonding (ARB) processes at 400 °C, thanks to which 24-layered composite materials were produced. The modification of the ARB process was based on the application of the rotational speed asymmetry (asymmetric accumulative roll bonding, AARB). It was adopted that the initial thickness of the composite stack amounted to 3 mm (1 mm for each composite). The rolling was done in the laboratory duo D150 rolling mill with the application of the roll rotational speed asymmetry and symmetry a_v_ = 1.0 (ARB) and a_v_ = 1.25 and 1.5 (AARB). In this manuscript, it was proved that introducing the asymmetry into the ARB process for the tested Al/Mg/Al composite has an impact on the activation of additional shear bands, which results in higher fragmentation of the structure in comparison to the symmetrical process. Due to the application of the AARB, the reduction of the grain size by 17% was obtained, in comparison to the conventional ARB. Not to mention that at the same time there was an increase in strength of the fabricated multilayered composite.

## 1. Introduction

Growing demand for light structural materials characterized by high strength-to-weight ratio, especially in applications where weight is an important factor that has an influence on finished products, has resulted in an increasing interest in magnesium alloys in recent years [1]. In aviation, automotive and space industries, light metal alloys have grown in importance. These light metal alloys include aluminium and titanium alloys as well as the composites based on them, which is associated with new fabrication methods. One of the main factors which plays a key role in determining mechanical properties for almost all crystalline materials is an average grain size in the material applied for production. Both for ultrafine-grained (UFG) and fine-grained materials, a key factor for obtaining high strength properties is the use of unconventional plastic processing methods with high strain—severe plastic deformation (SPD) [2,3,4]. Currently, the most common SPD processes are as follows: high-pressure torsion (HPT) [5,6], equal-channel angular pressing (ECAP) [7,8], accumulative clad rolling (ACR) [9,10], accumulative roll bonding (ARB) [11,12], asymmetric rolling (AR) [13,14] and asymmetric accumulative roll bonding (AARB) [15,16]. Currently, the most common SPD processes are as follows: high-pressure torsion (HPT) [5,6], equal-channel angular pressing (ECAP) [7,8], accumulative clad rolling (ACR) [9,10], accumulative roll bonding (ARB) [11,12], asymmetric rolling (AR) [13,14] and asymmetric accumulative roll bonding (AARB) [15,16]. 

Rolling asymmetry is defined by the use of asymmetry coefficient a_v_. Rolling asymmetry can be obtained by using rolls with different diameters (*D_u_ ≠ D_l_*) with the same rolling speed (v_u_ = v_l_) or with the rolling speed of upper and lower rolls (*v_u_ ≠ v_l_*) and the same diameters of the rolls (D_u_ = D_l_). The asymmetry value can be determined by the following Equation (1):(1)$$a_{v}\;=\;\frac{D_{u}}{D_{l}}\;=\;\frac{v_{u}}{v_{l}},$$
where: *D_u_, D_l_*—diameter of the upper and lower roll, *v_u_, v_l_*—rotational speed of the upper and lower roll.

Among the widely known plastic processing methods with high strain, methods based on ARB, AR, and AARB show the potential to be used in industry. The additional advantage of ARB and AARB processes, as compared to other SPD methods, is the possibility to utilize them in the continuous production of large-sized homogeneous and composite materials with the use of existing industrial designs [17]. The conventional ARB is based on repeated rolling of two or more sheets folded together with 50% rolling reduction, as a result of which individual layers are permanently bonded [18] using the friction forces between the components and diffusion phenomena. The interfacial bonding is a consequence of both the occurrence of a mechanical bonding on a macro-scale and the atomic bonding on a micro-scale [19,20]. To improve the quality of the workpieces, the bonded surfaces are cleaned of oxides and impurities and to improve the effectiveness of diffusion phenomena, the ARB process is run at elevated temperature and below the recrystallization temperature. The heat treatment is applied after the ARB process both as stress relief annealing and annealing that intensifies the diffusion phenomenon at the bonding interface of layers [21,22,23]. 

The manuscript [24] was dedicated to study the impact of ARB (6 cycles) on changes in the microstructure and mechanical properties of technical aluminium with different purity ranging from 99.2% up to 99.999% Al. It was concluded that intervals and boundary fraction with high angle is due to total uniform strain, taking into account the impact of the shear strain. For Al 99.2% and 99.99%, a nanometric structure dominated by high-angle boundaries was achieved. In the case of Al being 99.999%, recrystallization and growth of grains occurred, which caused the lack of nanostructure. The results of paper [25] revealed that ARB is a promising process for fabricating ultrafine-grained structures in aluminium sheets, and the average grain size after 5-cycle ARB reached approximately 300 nm. Meanwhile, a remarkable enhancement in the strength was achieved, and the value was about three times the strength of the initial material. In papers [26,27] one- and two-pass ARB processes of AZ31 alloy were analyzed, and it was demonstrated that due to rolling, significant fragmentation of the structure occurred, which in turn induced the improvement of strength properties. The results of the paper by [28] for AZ31 indicate that significant grain refinement is observed after the first two cycles at the highest ARB temperature (350 °C) as a result of dynamic recrystallization, which is necessary for the subsequent ARB cycles at a relatively lower temperature (250 °C) with the aim of restricting grain growth. No significant finer grain size was observed in the fifth and sixth cycles while the microstructure homogeneity is further improved. The grain structure can be effectively refined at lower ARB processing temperature and higher cycles.

The ARB was also applied to produce laminated metallic materials, which cannot be bonded with traditional methods [29]. In papers [30,31], the use of the ARB process to bond heat-treatable aluminium alloys (AA6061/AA7075/AA2219) with non-heat-treatable aluminium alloys (AA1050, AA5086) was elaborated. The applied rolling at elevated temperatures allowed for the production of multilayered aluminium composites of the ultrafine-grained structure with different properties. In recent years, the ARB started to be used for the fabrication of multilayered composites characterized by fine-grained structure, wherein individual layers are made of different metals such as Al/Cu [15], Al/Ni [16,32], Al/Zn [33], Ti/Al [34], Cu/Zr [35], Cu/Ni [36] and Mg/Al [37,38]. In the manuscripts quoted above, the conventional ARB method was used. In each case, the increase in strength properties after individual rolling cycles was demonstrated. The modification of the ARB is based on increasing the tangential stresses by introducing the rotational speed asymmetry into rolling to activate additional deformation mechanisms—AARB [39]. This modification was applied for the first time in [40] to obtain an ultrafine-grained structure of copper. In recent years, the majority of the investigations into the AARB have focused on using it in the fabrication of ultrafine-grained structures for pure copper [40,41], aluminium alloys [42] and magnesium alloys [43]. It has been shown that the introduction of additional roll rotational speed asymmetry into the ARB process results in faster grain refinement in fewer rolling cycles, as compared to the conventional ARB. The modified AARB process is also utilized to produce composites made of different metals [13,44,45,46]. Reference [44] proved that asymmetric accumulative roll bonding (AARB) of the aluminium-titanium (Al/Ti) composites leads to the increased plasticity thanks to the occurrence of shear strain. Additional shear strain exhibits a gradient across the thickness of the sample and increases initiation from the layer, which has contact with the lower speed roller to the roller rotating at a higher speed. In addition, deformation caused by shearing resulted in the increased grain refinement, and in consequence, grain size and hardness gradients. In [45] the AARB method was applied to fabricate the AA1050/AA7050 composite. The introduction of asymmetry had an influence on the unsteady flow of individual layers, which caused its waviness across the thickness of the rolled stack. This was due to the different properties of particular components. Initial investigations into the application of AARB for Mg/Al composites were provided by the authors in [13]. This paper described the results of experimental research of the rolling process via ARB or AARB of three-layered Al/Mg/Al sheets. The tests performed were limited to only one rolling cycle. The microhardness tests carried out for individual sheet layers processed in ARB and AARB did not demonstrate significant changes in the microhardness values for aluminium layers. However, it was detected that the microhardness of the Mg layer considerably increased during AARB cycles—from 72 HV (Vickers Hardness) up to 79 HV. The microstructure analysis of the Al/Mg/Al sheets indicates that good bonding between individual layers and grain refinement of Mg alloy had an impact on better strength properties of multilayered sheets fabricated in AARB, as compared to sheets produced via the ARB process.

The results presented in this manuscript were aimed at determining the influence of asymmetry value of the roll rotational speed on the structure and strength properties of Al/Mg/Al multilayered composite. The tests were conducted with the use of the traditional ARB and modified AARB process. Experimental tests were supplemented by numerical calculations with the use of a Finite Element Method FEM-based computer program. These tests adopted that the total primary thickness of the feedstock was 3 mm (Al/Mg/Al sheets with a thickness of 1 mm were used). Rolling was done in four cycles. In the first cycle, apart from the conventional ARB with a_v_ = 1.0, the roll rotational speed asymmetry (a_v_) was 1.25 and 1.5 (AARB). In the subsequent cycles, the rolling process was carried out via the conventional ARB.

## 2. Materials and Methods

To investigate the rolling process of the ARB- and AARB-fabricated multilayered composites, two types of materials were applied; technically pure aluminium grade 1050A and AZ31 magnesium alloy, the chemical compositions of which are shown in Table 1. 1050A aluminium samples were cut from the sheet with the thickness of 1 mm, and magnesium alloy samples were designed at the Czestochowa University of Technology (Poland). AZ31 alloy feedstock with the primary height of 70 mm and width of 150 mm was subjected to the rolling process in the industrial duo D300 rolling mill at 400 °C to a final thickness of 1 mm. To reduce internal stresses that remained after the rolling, the annealing process was performed. The annealing was conducted at 350 °C for 2 h. The samples made in this way were then cut and folded in a 25 mm × 3 mm × 150 mm stack, wherein the primary thickness of individual layers was 1 mm (1 mm + 1 mm + 1 mm) [46]. After the samples were folded in stacks, they were heated in a chamber furnace to 400 °C and annealed by 10 min before each rolling cycle. 

To investigate the rolling process of Al/Mg/Al multilayered composites, a two-high rolling mill (Czestochowa University of Technology, Czestochowa, Poland) was used with the nominal roll diameter of 150 mm and the length of roll barrel of 170 mm. Each roll had a separate drive from asynchronous alternating current (AC) motor with the nominal power of 7.5 kW, through the reduction gear with a velocity ratio of 1:22.4 and a transmission shaft. It enabled to introduce roll circumferential speed asymmetry. The engine was controlled by the frequency converter ACS-601 manufactured by ABB Industry (Västerås, Sweden). A diagram of the laboratory stand is illustrated in Figure 1.

Rolling processes in ARB and AARB of the Al/Mg/Al multilayered composite were conducted in three variants, but the rotational speed asymmetry was introduced only into the first rolling cycle (variant I—symmetric process a_v_ = 1.0; variant II—asymmetry a_v_ = 1.25; variant III—asymmetry a_v_ = 1.5). Table 2 contains the pass schedule for conducted research and height measurements of Al/Mg/Al multilayered composite after each of 4 rolling cycles. It also includes the applied rolling variants, classic ARB and modified AARB, with the calculated values of rolling reductions and true strains. 

Introducing the asymmetrical rolling into the subsequent cycles induced the propagation of edge transverse cracks across the whole thickness of the workpiece. Due to this, no asymmetric accumulative roll bonding (AARB) was performed in the next cycles. The subsequent cycles were then carried out with the same roll rotational speed via the ARB process, which considerably reduced the crack propagation and formation. In the next three cycles, the process was conducted symmetrically for all variants [46].

The metallographic examination was used to determine the initial structure of materials applied for research (Figure 2). The revealed structure is typical for plastically pre-deformed materials. In AZ31 alloy, groups of small grains and bigger grains can be seen. The average grain size of the tested materials before the rolling process was determined by the use of random secants. The initial grain size of AZ31 magnesium alloy was 25 μm, and of 1050A aluminium was 107 μm. The mechanical properties of the particular materials used in tests are presented in Table 3.

The 1050A/AZ31/1050A multilayered materials were cut parallel to the rolling direction and prepared for microscopic observations according to standard metallographic procedures. The final polishing was performed using 0.05 μm colloidal silica suspension. Metallographic examinations of Al/Mg/Al samples were undertaken after individual rolling cycles. The microstructure of AZ31 alloy was revealed with the use of a reagent. The chemical composition of it was as follows: 20 mL ethanol + 2 mL acetic acid + 2 g picric acid; and the aluminium was etched with 5% HF (hydrofluoric acid). The microstructure of the bonding zone of the Al/Mg/Al materials was determined by means of a Nikon ECLIPSE MA200 optical microscope (Nikon Imaging Japan Inc., Tokyo, Japan). Microhardness tests of Al/Mg/Al multilayered composites obtained after subsequent passes of ARB and AARB processes were performed using a FM-700 FutureTech microhardness tester (Future-Tech Corp., Fujisaki, Japan). Five microhardness tests were employed for each layer of the multilayered sample. This was done by pressing the Vickers indenter into the test material. HV microhardness measurement was taken with the load of 50 g. Tests of the strength were undertaken by using Zwick Z100 machine (ZwickRoell, Ulm, Germany). The experimental research was supplemented with computer simulations executed in the FEM-based Forge2011^®^ (Transvalor, Sophia Antipolis, France) computer software.

To properly do the theoretical tests of the analysis of the ARB and AARB rolling process of the Al/Mg/Al multilayered composite, it was necessary to determine the relationship between the flow stress and the basic parameters of plastic processing—temperature and strain rate. The flow stress was defined in a plastometric hot compression test for plane strain state, which corresponds to the rolling process of metal plates. Plastometric tests were carried out in the Gleeble 3800 system (version 3800, Dynamic Systems, Poestenkill, NY, USA), which is a simulator of metallurgical processes. Compression tests were done at a constant temperature of a deformed material in a vacuum chamber. In plastometric tests, rectangular-shaped samples (10 mm × 15 mm × 20 mm) were used. The conducted tests agreed with the parameters illustrated in Table 4 [47].

Based on the test results, approximation factors were developed for the equation of the flow stress of tested materials. To approximate the calculations of plastometric tests, the Hensel–Spittel function was used [48], which is expressed by the following Equation (2):(2)$$\sigma_{f}=A_{1}e^{m_{1}T}\varepsilon^{m_{2}}e^{\varepsilon m_{4}}{\dot{\varepsilon }}^{m_{3}}{\left(1+\varepsilon \right)}^{\frac{m_{5}}{\varepsilon }}e^{\varepsilon m_{7}}{\dot{\varepsilon }}^{m_{8}T}T^{m_{9}},$$
where: *σ_f_*—flow stress [MPa], *T*—temperature [°C], *ε*—true strain, $\dot{\varepsilon }$—strain rate [s^–1^], *A*_1_, *m*_1_÷*m*_9_—coefficient of function (1).

The approximation factors of the flow stress Equation (1) were shown in Table 5. These factors were then entered into Forge2011^®^ computer program. The thermo-mechanical simulation of the Al/Mg/Al multilayered materials was carried out with the use of a visco-plastic model in the triaxial state of strain by using the Forge2011^®^ program, whereas the properties of the deformed material were described according to the Norton–Hoff [49,50] conservation law written in form (3):(3)$$S_{ij}=2K\left(T,\dot{\bar{\varepsilon }},\bar{\varepsilon }\right){\left(\sqrt{3}\dot{\bar{\varepsilon }}\right)}^{m-1}{\dot{\varepsilon }}_{ij},$$
where: *S_ij_* is the deviatoric stress tensor, $\dot{\bar{\varepsilon }}$ is the equivalent strain rate, *ε_ij_* is the equivalent strain rate tensor, $\bar{\varepsilon }$—equivalent plastic strain, *T* is the temperature, *K*—consistence being dependent on the flow stress *σ_f_*, *m*—factor which characterizes hot metal deformation (0 < *m* < 1).

Figure 3 shows examples of sample plastometric testing results and plastic flow stress–strain curves and obtained from their approximation for the materials under investigation. The solid lines with empty markers denote the curves representing the plastometric testing results, while the solid lines with filled markers denote the curves obtained from the approximation of the plastometric testing results.

To perform numerical calculations of ARB and AARB, it was indispensable to design spatial models of the band shape and three-dimensional models of rolls with auxiliary tools (Figure 4). The tools model was constructed from a triangle-based mesh, while the Al/Mg/Al feedstock model was constructed from tetrahedral elements. The Al/Mg/Al model with dimensions (3 × 1 mm—height; 2 mm—width and 60 mm—length) was built from ca. 3 × 40,000 elements for each layer, each element of the average edge length of 0.3 mm. In order to increase the speed of calculations in the simulation project, two parallel planes limiting the spreading were introduced for the deformed material, which enabled us to reduce the number of elements for the band (2 mm), Figure 4. 

Using the results of previous research [46], where it was demonstrated that already in the first pass there is a full bonding between particular layers of the multilayered material, the bonding between individual layers was defined as closely adjacent in numerical investigations. Thus, compliant with reference [51], in numerical modeling of the ARB and AARB processes, the friction factor between individual layers was assumed to be 1. The general conditions of rolling simulation that were adopted agreed with the parameters of the D150 mm laboratory rolling mill, wherein experimental tests were conducted. In addition, the appropriate boundary and initial conditions were assumed: the temperature of the multilayered charge was 400 °C (both for outer layers of aluminium alloys and the inner layer of magnesium alloy), tool temperature (T_tool_) was 60 °C; ambient temperature (T_air_) was 20 °C; heat exchange coefficient between the band and the rolls (α) was 3000 [W/(m^2^K)]; heat exchange coefficient between aluminium and magnesium layers (α_b_) was 2000 [W/(m^2^K)]; heat exchange coefficient between the band and the environment (α_air_) was 10 [W/(m^2^K)]. Numerical calculations were made with the use of the mixed Coulomb–Tresca friction model. The value of the friction coefficient was assumed to be 0.35; and the friction factor was adopted to be 0.7 based on [52]. The bonding between particular layers was defined as closely adjacent. Nodes of particular lattices were not shared. The rolling speed amounted to 0.2 m/s (for conventional ARB process); the following coefficients of roll rotational speed asymmetry for AARB process a_v_ = 1.25 and 1.5 were adopted. Rolling reduction amounted to 50%. This paper focused on the numerical modeling of the first rolling cycle (pass) where, in experimental tests, the rolling process was performed for four rolling cycles (passes) according to the variants described above.

## 3. Results and Discussion

### 3.1. Analysis of the Numerical Modeling Results

The first part of the tests aimed to determine the impact of the asymmetry on the activation intensity of additional shear bands in the Al/Mg/Al multilayered composite, which was fabricated via the ARB and AARB process. To explain the plastic flow of the Al/Mg/Al multilayered composite, the analysis of the component (v_z_) of the plastic flow rate was conducted in the rolling direction (Figure 5).

The distribution of the flow rate (v_z_) obtained for the symmetric process is a typical distribution for the rolling process of plane products. The isolines are of a symmetric character in relation to the horizontal longitudinal symmetry axis of the band. In the band subjected to the rolling process, the forward and backward slip zones can be distinguished. The distribution of the v_z_ component of the plastic flow illustrated in Figure 5a enables the band to exit the roll gap as being straight. The results of the numerical calculations on the character of the plastic flow of individual layers of the multilayered band are consistent with the results of the manuscript [53,54]. The application of the roll rotational speed asymmetry disturbs the uniform distribution of the v_z_ component, which is particularly visible in the forward slip zone and neutral plane zone, wherein the v_z_ flow rate is equal to the rotational speed of rolls. The non-uniform distribution of the v_z_ component of the flow rate for the band area marked with broken lines (Figure 5b) affects the distribution of the τ_yz_ shear stresses. Increasing the roll rotational speed asymmetric factor substantially intensified the disturbance of the symmetric distribution of isolines in relation to the v_z_ component (Figure 5c). It is particularly noticeable in the forward slip and neutral zone. Moreover, attention should also be paid to the flow rate value relative to the horizontal longitudinal symmetry axis of the band. Due to the introduced symmetry, an increase was noted in the flow rate of the multilayered band in the forward slip zone by 15% as compared to the symmetric rolling. More non-uniform distribution of v_z_ component of the flow rate for the band area marked with broken lines (Figure 5c) has an impact on the distribution of tangential stresses (Figure 6).

One of the factors determining the fragmentation of the structure in the deformed band and bonding quality of individual layers is tangential stresses, which have an influence on the activation of additional shear strains. Figure 6 presents the distribution of τ_yz_ tangential stresses along the longitudinal intersection of the Al/Mg/Al multilayered composite. During the symmetric ARB process (Figure 6a), the uniform distribution of τ_yz_ stresses in relation to the horizontal longitudinal symmetry axis of the band was recorded along the whole length of the roll gap. However, it should be mentioned that τ_yz_ stresses above and below this axis had the same values, but differed in signs. Figure 6b shows the distribution of τ_yz_ stresses for the asymmetrical rolling process for a_v_ being 1.25. In this case, we can notice a different distribution of τ_yz_ stresses from that achieved via the ARB process (Figure 6a). Band section area indicated by broken lines in Figure 6b was negatively influenced by τ_yz_ tangential stresses. Particularly important is the occurrence of τ_yz_ stresses on the intersection of the magnesium layer and bonding areas between individual layers. Due to the introduction of the roll rotational speed asymmetry, the occurrence of tangential stresses is recorded in a plane oriented perpendicular to the rolling direction. These stresses can be seen in all layers of the workpiece along its entire height. The biggest stress area with the positive sign exists in the lower aluminium layer from the roll side with the lower rotational speed, and the smallest stress area can be observed in the upper aluminium layer from the roll side with the higher rotational speed. Figure 6c outlines the distribution of τ_yz_ stresses for the asymmetric rolling process for a_v_ equal to 1.5. In this case, the similar character of the stress distribution is noted as in the variant above illustrated in Figure 6b (a_v_ = 1.25). The band intersection area demonstrated with broken lines (Figure 6c) is under the influence of τ_yz_ stresses with the negative sign. It is of paramount importance to increase the length of the τ_yz_ stress area in comparison to the symmetrical rolling, and with the lower asymmetry value across the longitudinal intersection of the magnesium layer, and in the bonding area between individual layers as well. Increasing the value of the asymmetry coefficient impacted not only the increased length of τ_yz_ stresses on the longitudinal section of the multilayered material, but it also increased their value. Thus, it can be concluded that introducing the rotational speed asymmetry into the rolling process of Al/Mg/Al multilayered composite machined via the ARB process has an impact on the formation of additional τ_yz_ stresses. Due to this, additional shear bands can be activated and the value of the predefined a_v_ coefficient has an influence on the length of the area, where they exist (Figure 6). The occurrence of additional τ_yz_ stresses, as compared to the symmetrical rolling via the ARB, should induce faster fragmentation of the structure.

### 3.2. Analysis of the Experimental Results

Samples for metallographic examinations were taken from the ARB- and AARB-fabricated Al/Mg/Al multilayered composite after each rolling cycle (Figure 7) in such a way that the cutting plane of the sample was perpendicular to the rolling direction, on which the appropriate etching was performed that enabled structural tests to be carried out. This paper is limited to the microstructure analysis after the first rolling cycle, which corresponded to the numerical simulations. It was also focused on the microstructure analysis after the last (fourth) rolling cycle. The images of metallographic microsections of test samples after the first and the fourth rolling cycles are provided in Figure 8, Figure 9, Figure 10, Figure 11, Figure 12 and Figure 13.

Based on Figure 8b, it can be concluded that no intermetallic phases after the first rolling cycle between the Mg–Al layers were observed. Also, it can be noticed that the shear bands formed in the layer of magnesium alloy (Figure 8c,d). They are oriented along the rolling direction as a result of the classic ARB process which is characterized by the rotational speed symmetry of cooperating rolls. Shear bands are directed towards the centre of the band and are arranged in the so-called herringbone pattern. In the aluminium layer (Figure 8e), large highly elongated grains are oriented along the rolling direction. 

Figure 9 shows the Al/Mg/Al multilayered composite after the first rolling cycle for the second rolling variant via AARB. The predefined coefficient value of the roll speed asymmetry (a_v_) was 1.25. In the multilayered sample rolled via AARB with a_v_ = 1.25, no intermetallic phases between Mg–Al were recorded (Figure 9b). In comparison to the ARB symmetrical process, more intensive deformation of the layers of AZ31 alloys is detected. Due to the asymmetry introduced into the rolling process, crack initiation and discontinuities in shear bands illustrated in Figure 9c,d is observed. As a result of the activation of shear bands, much greater grain refinement is registered within them as compared to the symmetric ARB process. In the aluminium layer (Figure 9e), due to the introduced asymmetry, highly elongated and more refined grains oriented towards the rolling direction are noticed.

Figure 10 presents Al/Mg/Al multilayered composite after the third rolling variant via AARB. The predefined coefficient value of roll rotational speed asymmetry was a_v_ = 1.5. The metallographic examinations (Figure 10a,b) that were performed for Al/Mg/Al multilayered composite machined via the AARB process for a_v_ = 1.5 in the first rolling cycle did not demonstrate the occurrence of the intermetallic phase at Mg–Al. However, increasing a_v_ up to 1.5 resulted in considerable densification of shear bands and cracks in the centre of the AZ31 layer (Figure 10c,d). It is crucial to bear in mind that the densification of shear bands in the middle layer of magnesium alloy resulted in the increased grain refinement in its entire volume. As in the case of a_v_ = 1.25 asymmetry, here, highly elongated aluminium grains directed along the rolling process are also visible.

Figure 11, Figure 12 and Figure 13 outline the microstructure after the fourth ARB rolling cycle for samples rolled according to variants I, II and III, respectively, which were fabricated via AARB in the first rolling cycle. After the fourth rolling cycle, the composite consisted of 24 layers. It can be concluded that for all the analyzed variants, there was a fragmentation of AZ31 alloy in the 1050A aluminium matrix. Such a system of interfering layers can be called a composite, in the structure of which Al matrix contains fragments of AZ31 alloy elongated towards the rolling direction. After the fourth rolling cycle in the AZ31 alloy, fine grain is detected with few shear bands (Figure 11d). Aluminium layers in the analyzed rolling cycle are distinguished by elongated grains along the rolling direction (Figure 11e). The layer of intermetallic phases is highly fragmented (Figure 11b,c).

Figure 12 shows Al/Mg/Al multilayered composite after the fourth rolling cycle (a_v_ in the first pass was 1.25). The rolling process with a_v_, which was 1.25 in the first pass, influenced the increase of the fragmentation in the magnesium alloy and elongation of fragmented layers along the rolling direction (Figure 12a). The formation of the intermetallic phase was also discovered at the bonding edge (Figure 12b,c). After rolling in the analyzed areas of magnesium alloy (Figure 12d), accumulation of strong deformations caused by densification of shear bands was observed. The aluminium matrix is characterized by highly elongated grains oriented towards the rolling direction (Figure 12e).

A multilayered test sample machined according to variant III with a_v_ in the first pass equal to 1.5 (Figure 13a) was distinguished by a higher proportion of fragmented AZ31 aluminium layers as compared to variant I and II. The examined areas of Mg alloy illustrated in Figure 13d are also represented by high accumulation of deformation caused by the densification of shear bands. It should also be mentioned that the aluminium matrix consists of highly elongated grains oriented towards the rolling direction (Figure 13e).

The fragmented layers of magnesium alloy present in the material are formed as a result of the rupture (cracking) of this layer due to high strain (shear stresses) and filling this space with aluminium. Subsequent rolling cycles (passes) will trigger the formation of composite material with the increasing discontinuities in magnesium layers. The variation of the cracks at different cycles of the ARB processed Al/Mg compound alloy were explained in detail in [38]. In the present work, the morphology and character of the cracks in the Mg/Al interface are in a good agreement with the quoted paper.

Figure 14 presents the grain sizes determined for the AZ31 alloy and 1050A alloy after subsequent cycles for the rolling variants examined. Based on the results obtained, a slight impact of the rotational roll speed asymmetry introduced into the ARB process on the aluminium grain size can be observed, where the difference in size during the first rolling cycle between variant I and variant III was 2%. Introducing asymmetry into the first cycle transfers to the other rolling cycles, causing a slight difference in the grain size of the aluminium layer. It should be noted also that for the layer made of the AZ31 alloy, the highest grain refinement was obtained after the first rolling cycle, irrespective of the applied rolling variant. Based on the accumulated results, it can be concluded that the introduction of roll rotational speed asymmetry into ARB significantly affected the reduction of grain size. The implementation of asymmetry a_v_ = 1.25 into the AARB process resulted in the activation of additional shear bands, which contributed to grain refinement in the area where they exist. A further increase in asymmetry up to a_v_ = 1.5 has led to even higher densification of shear bands. In the next cycles, such a significant fragmentation of the microstructure of AZ31 alloy is not noticed. However, it should be taken into consideration that the impact of asymmetry on the size of the obtained microstructure of magnesium alloy in the Al/Mg/Al multilayered composite remained after each of the analyzed rolling cycles.

In order to determine the influence of the AARB process on mechanical properties of multilayered materials, tensile and microhardness (HV) tests are performed. Figure 15 shows the values of the ultimate tensile strength (UTS) of samples obtained after individual rolling cycles.

Building upon the data exhibited in Figure 15, it can be deduced that after the first rolling cycle of Al/Mg/Al multilayered composite, the highest strength was recorded for specimens rolled via the conventional ARB process, compared to those fabricated via the AARB. In the first rolling cycle, the implementation of roll rotational speed asymmetry into the ARB process induced the occurrence of cracks in the AZ31 layer, which resulted in the reduction in strength of the samples rolled according to variant II and III. In the second rolling cycle, the Mg layer experienced cracks in the samples rolled according to the I variant. These cracks resulted in a decrease in strength properties. For test specimens machined via AARB for variants II and III, wherein due to the introduction of asymmetry into the first cycle there was a greatest reduction of grain size (Figure 14), an increase in values of the UTS takes place. In the subsequent rolling cycles, a delayed effect of the application of the AARB in the first rolling cycle for II and III variant can be observed. It results in the higher increase in the UTS for samples rolled via AARB as compared to ARB, which was due to the increased fragmentation of the microstructure. The results of microhardness tests were exhibited in Figure 16. The initial microhardness values were 26.2 HV for aluminium and 68.8 HV for AZ31 alloy.

Based on the data in Figure 16, it can be deduced that for each layer of Al/Mg/Al composite, microhardness value by the indenter’s load of 50g increases linearly with the number of passes. In relation to the initial material of 1050A aluminium, the lowest value was obtained in the first rolling cycle, and the highest in the fourth rolling cycle due to rolling via the classic ARB. The same character of changes in microhardness measurements of Al/Mg/Al multilayered composite was observed while rolling in the first cycle via AARB with a_v_ = 1.25 and a_v_ = 1.5. It should be noted, however, that in spite of the same character of changes in microhardness, in the last rolling cycle, an increase of approx. 5.5% occurred between ARB and AARB with a_v_=1.5. In the layers of AZ31 magnesium alloy investigated, a linear increase in microhardness was detected in subsequent passes, as was found in 1050A aluminium. However, in this case, greater diversification of the microhardness values obtained was observed between the classic ARB rolling process and the modified AARB with a_v_ = 1.5. The highest difference was noted during the first rolling cycle, and it amounted to nearly 8%. After the fourth rolling cycle, the difference in microhardness of the layer of magnesium alloy between ARB and AARB with a_v_ = 1.5 was only 1.5%, which may be indicative of the substantial impact of asymmetry (first rolling cycle) on the microhardness of individual layers. In particular, it can be observed after the first rolling cycle for the layer of magnesium alloy. In the next passes, smaller differences in the obtained microhardness values for ARB and AARB processes were found (rolling in the next cycles only via ARB). 

## 4. Conclusions

Based on the performed tests, it can be concluded that the application of ARB and AARB enables the formation of Al/Mg/Al multilayered composite. The results of theoretical and experimental investigations may give rise to further use of the examined process in industrial conditions. Moreover, given the analysis of the results, it can be concluded that: (1)favourable impact of the introduction of the ARB into the roll rotational speed asymmetry increases tangential stresses τ_yz_, which was proved in experimental tests;(2)the application of roll rotational speed asymmetry in the first rolling cycle has a beneficial effect on the activation of additional shear bands, especially in magnesium alloy, which was demonstrated in numerical calculations and microstructural examinations. It results in increased grain refinement as compared to the ARB process. The values of the asymmetry introduced into the ARB contributed to the decrease by 17% of grain dimension in the last rolling cycle of the AZ31 alloy relative to the ARB;(3)the suggested modification of the ARB caused an increase in microhardness of each layer and an improvement of strength properties of the Al/Mg/Al multilayered composite subjected to rolling;(4)the results of strength tests obtained revealed that the introduction of rolling speed asymmetry increases the tensile strength of the fabricated Al/Mg/Al composite, as compared to the composite obtained via the classic ARB process;(5)after the last rolling cycle, high fragmentation of the magnesium alloy layer is observed in the finished Al/Mg/Al composite. This increases along with the growth of the asymmetry coefficient in the first rolling cycle. The introduction of too high a roll circumferential speed asymmetry has an impact on the formation of a higher number of cracks.

## Figures and Tables

**Figure 1 materials-13-05401-f001:**
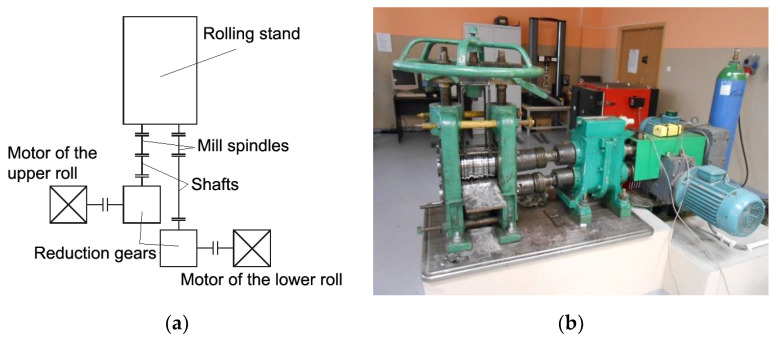
The layout of the D150 two-high rolling mill—(**a**) and a general view—(**b**).

**Figure 2 materials-13-05401-f002:**
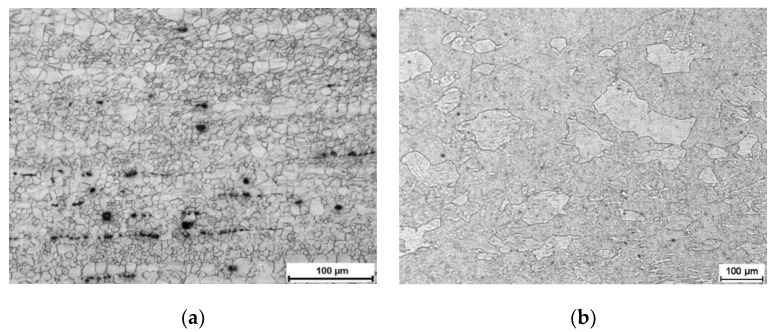
Initial microstructure of materials used in tests: (**a**) AZ31 magnesium alloy, (**b**) 1050A aluminium.

**Figure 3 materials-13-05401-f003:**
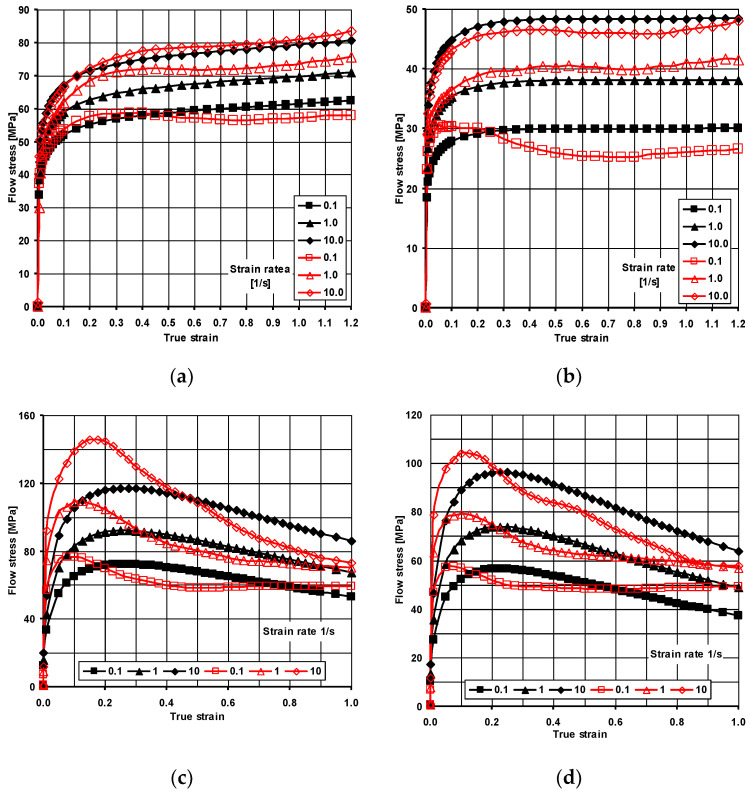
Flow stress curves of materials used in tests: (**a**) 1050A, 350 °C, (**b**) 1050A, 400 °C, (**c**) AZ31 alloy, 350 °C, (**d**) AZ31 alloy, 400 °C [47].

**Figure 4 materials-13-05401-f004:**
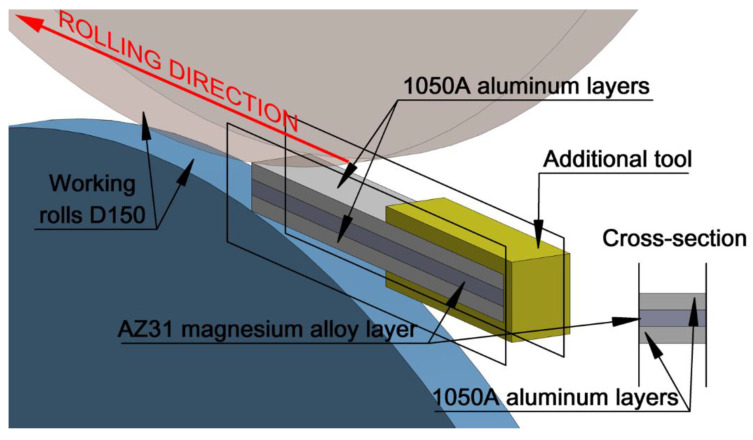
A schedule of the computer simulation design of the band rolling process of Al/Mg/Al multilayered composite.

**Figure 5 materials-13-05401-f005:**
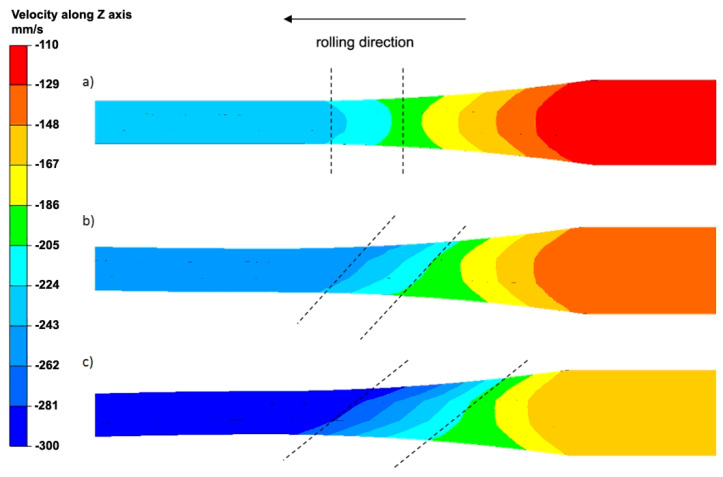
Distribution of the v_z_ longitudinal metal flow velocity component during the first rolling cycle: (**a**) a_v_ = 1.0 (ARB), (**b**) a_v_ = 1.25 (AARB), (**c**) a_v_ = 1.5 (AARB).

**Figure 6 materials-13-05401-f006:**
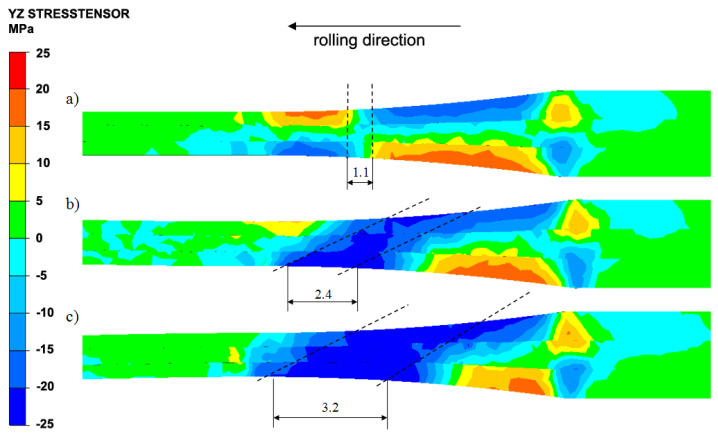
Distribution of the τ_yz_ tangential stress component during the first rolling cycle: (**a**) a_v_ = 1.0 (ARB), (**b**) a_v_ = 1.25 (AARB), (**c**) a_v_ = 1.5 (AARB).

**Figure 7 materials-13-05401-f007:**
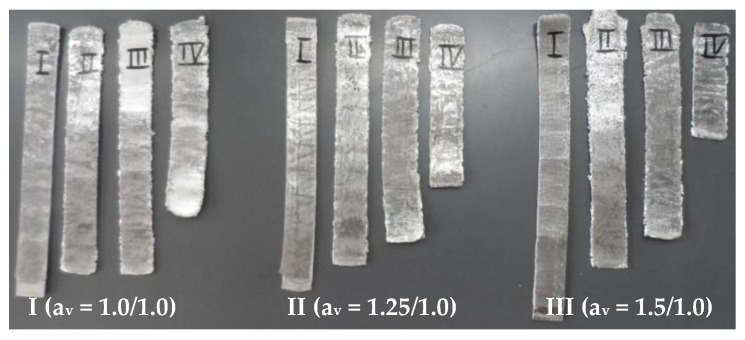
View of the Al/Mg/Al multilayered samples after particular cycles.

**Figure 8 materials-13-05401-f008:**
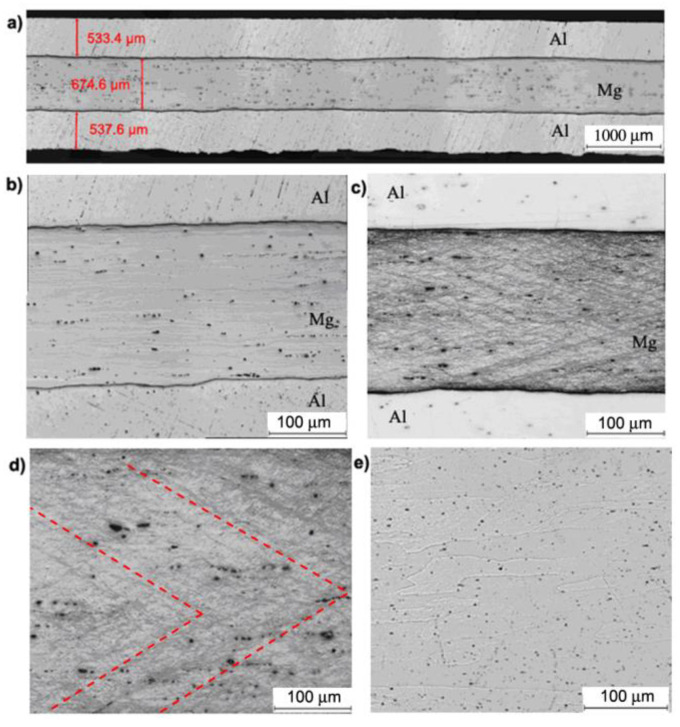
Al/Mg/Al multilayered sample after the 1st rolling cycle rolled acc. to variant I: (**a**,**b**) non-etched microsection, (**c**) etched microsection, (**d**) etched Mg microsection, (**e**) etched Al microsection.

**Figure 9 materials-13-05401-f009:**
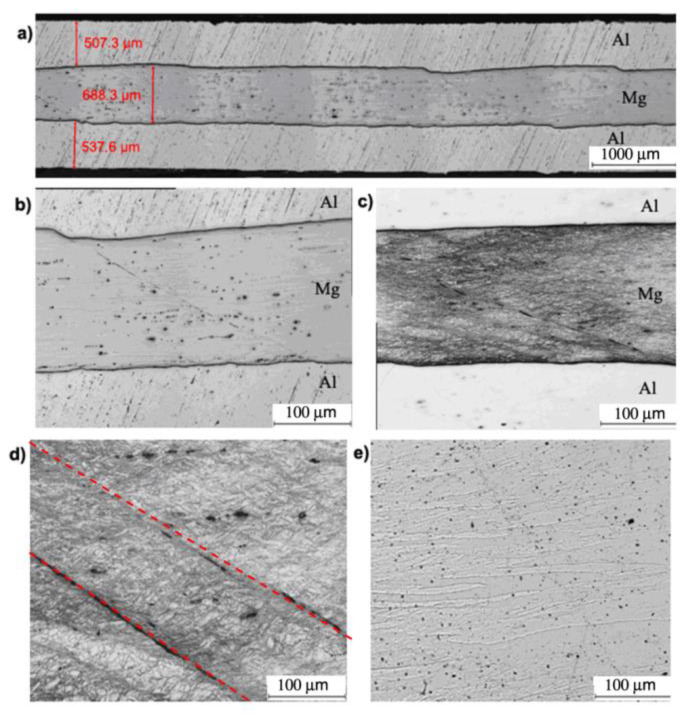
Al/Mg/Al multilayered sample after the 1st rolling cycle rolled acc. to variant II: (**a**,**b**) non-etched microsection, (**c**) etched microsection, (**d**) etched Mg microsection, (**e**) etched Al microsection.

**Figure 10 materials-13-05401-f010:**
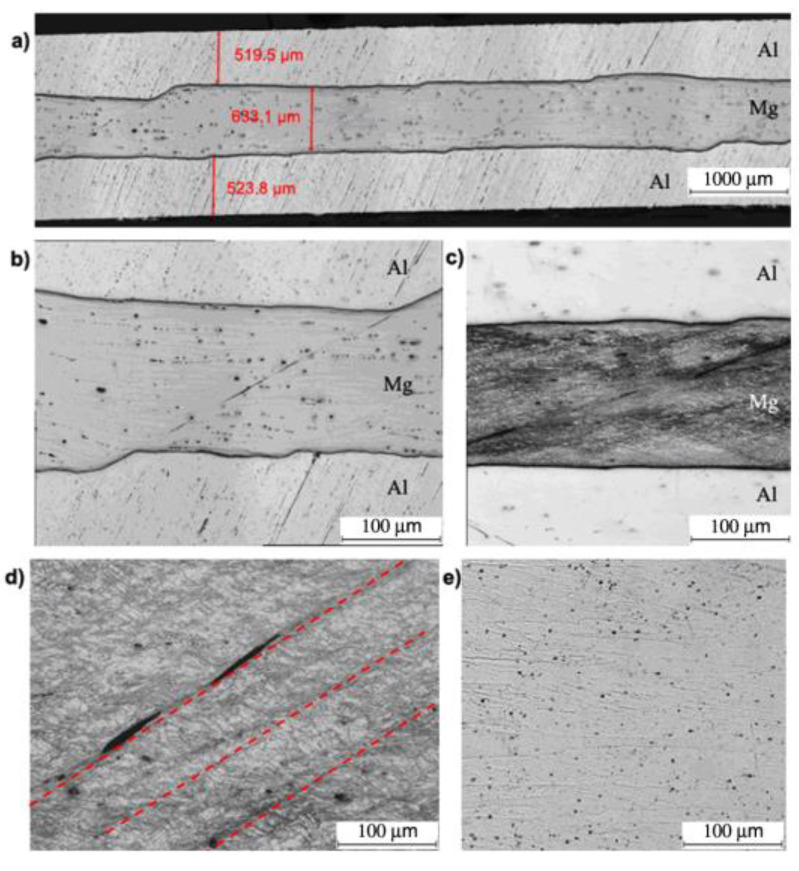
Al/Mg/Al multilayered sample after the 1st rolling cycle rolled acc. to variant III: (**a**,**b**) non-etched microsection, (**c**) etched microsection, (**d**) etched Mg microsection, (**e**) etched Al microsection.

**Figure 11 materials-13-05401-f011:**
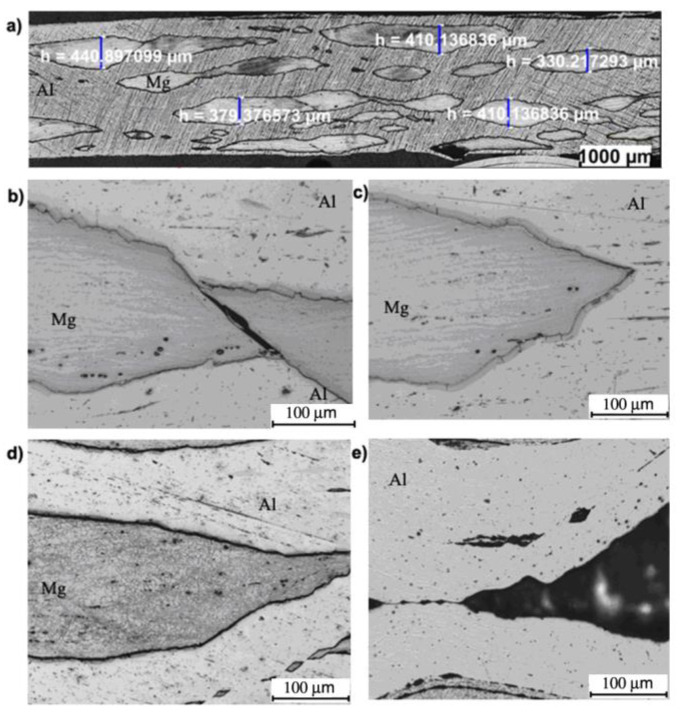
Al/Mg/Al multilayered sample after the 4th rolling cycle rolled acc. to variant I: (**a**,**b**) non-etched microsection, (**c**) etched microsection, (**d**) etched Mg microsection, (**e**) etched Al microsection.

**Figure 12 materials-13-05401-f012:**
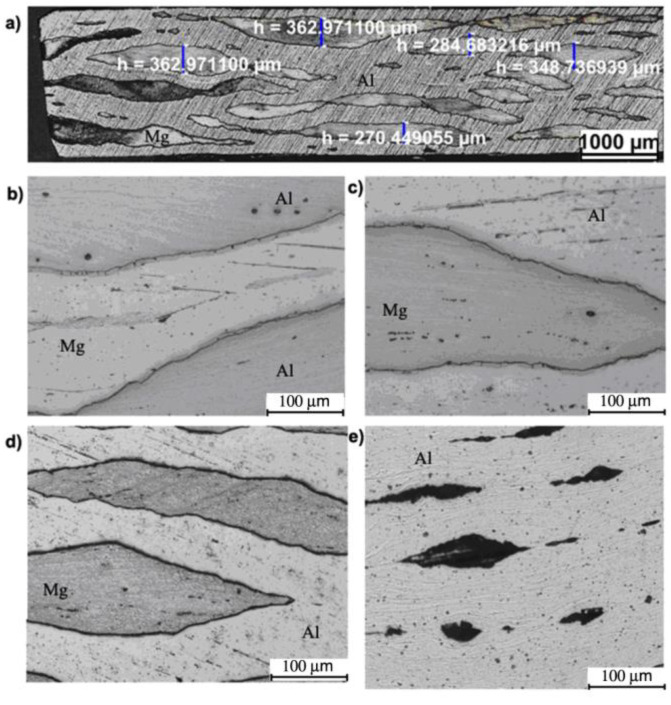
Al/Mg/Al multilayered sample after the 4th rolling cycle rolled acc. to variant II: (**a**,**b**) non-etched microsection, (**c**) etched microsection, (**d**) etched Mg microsection, (**e**) etched Al microsection.

**Figure 13 materials-13-05401-f013:**
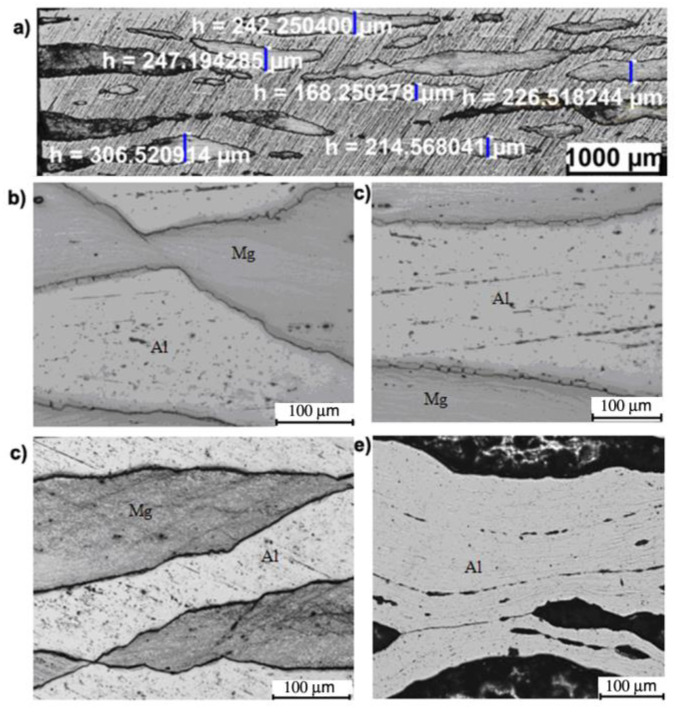
Al/Mg/Al multilayered sample after the 4th rolling cycle rolled acc. to variant III: (**a**,**b**) non-etched microsection, (**c**) etched microsection, (**d**) etched Mg microsection, (**e**) etched Al microsection.

**Figure 14 materials-13-05401-f014:**
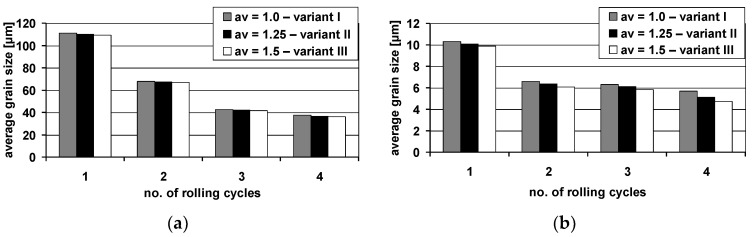
The average grain size of 1050A aluminium—(**a**) and AZ31 magnesium alloy—(**b**) after particular rolling cycles.

**Figure 15 materials-13-05401-f015:**
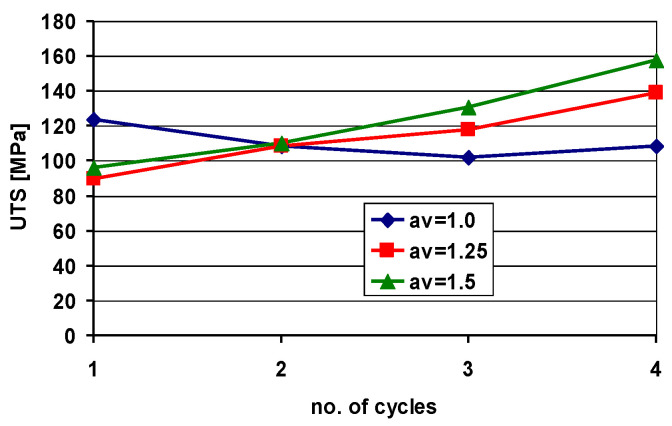
Ultimate tensile strength (UTS) for Al/Mg/Al multilayered composite fabricated in the ARB and the AARB processes.

**Figure 16 materials-13-05401-f016:**
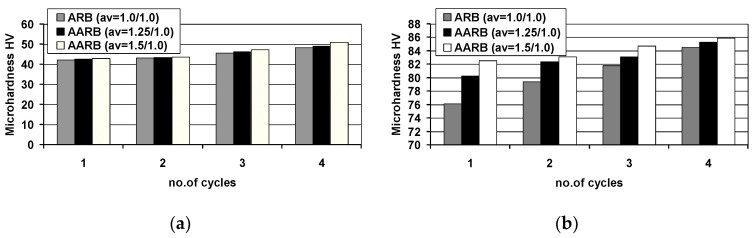
The average value of microhardness HV: (**a**) 1050A aluminium, (**b**) AZ31 magnesium alloy after particular rolling cycles.

**Table 1 materials-13-05401-t001:** Chemical composition of the alloys used in accumulative roll bonding (ARB) and asymmetric accumulative roll bonding (AARB) processes (wt. [%]).

Material	Al	Mg	Fe	Mn	Ni	Si	Zn	Cu
1050A	99.50	0.047	0.32	0.005	0.01	0.06	0.008	0.05
AZ31	3.5	95.0	0.01	0.4	0.01	0.1	0.8	0.05

**Table 2 materials-13-05401-t002:** Pass schedule and thickness reductions.

Method	Asymmetry Coefficient, a_v_	Cycle no.	Initial Thickness [mm]	Final Thickness [mm]	Reduction [%]	True Strain	
ARB	1.0	1	3.00	1.57	47	0.64	2.75
		2	3.14	1.49	52	0.74	
		3	2.98	1.52	48	0.67	
		4	3.04	1.51	50	0.70	
AARB	1.25	1	3.00	1.56	48	0.65	2.76
	1.0	2	3.12	1.47	53	0.75	
		3	2.94	1.50	49	0.67	
		4	3.00	1.50	50	0.69	
	1.5	1	3.00	1.56	48	0.65	2.77
	1.0	2	3.12	1.47	53	0.75	
		3	2.95	1.50	49	0.68	
		4	3.00	1.50	50	0.69	

**Table 3 materials-13-05401-t003:** Mechanical properties of Al 1050A and AZ31 magnesium alloy.

Material	Yield Strength, YS, [MPa]	Ultimate Tensile Strength, UTS [MPa]	Elongation A_10_ [%]
AZ31	130	240	10
1050A	30	80	40

**Table 4 materials-13-05401-t004:** Parameters used for plastometric tests.

Material	Temperature [°C]	Strain Rate [s^−1^]	True Strain [–]
1050A	350, 400, 450	0.1; 1.0; 10	up to 1.2
AZ31	350, 400, 450	0.1; 1.0; 10	up to 1.0

**Table 5 materials-13-05401-t005:** Coefficient of function (2).

Material	A_1_	m_1_	m_2_	m_3_	m_4_	m_5_	m_7_	m_8_	m_9_
1050A	0.08743	−0.0099	0.11325	−0.08845	−0.00058	−0.00153	0.196267	0.00048	1.71527
AZ31	0.68478	−0.0072	0.34242	0.02864	−0.08199	−0.00023	−0.00439	0.00022	1.41094

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
