# Peer review of "Effect of Asymmetric Accumulative Roll-Bonding process on the Microstructure and Strength Evolution of the AA1050/AZ31/AA1050 Multilayered Composite Materials"

_materials, 2020, doi:10.3390/ma13235401_

Round 1

Reviewer 1 Report

The paper presents an interesting study on evolution of microstructure and strength of a roll-bonded Al/Mg/Al multilayered composite. The fabrication of a light material is here in focus. Flow stress curves at different strain rates and microstructural changes over the rolling cycles are used to simulate and discuss the results.

Just a few comments and suggestions to improve the quality of the paper even further:

- a comment on the specific strength compared to other composites (not multilayer)

- the interface is mentioned, but not further used in the discussion, you only mention that the character of the cracks are in an good agreement to paper [38] on page 13 line 343… please check here the reference [38] …it is on pure copper and not Al/Mg

- I recommend to move results (Figure 1 and Figure) in chapter 3

- Table 2: might it be better to write …up to 1.2…

- can you add, why you used different true strain values…

- Figure 12: please prepare for black and white printing… maybe use different texture in the bars

Author Response

Dear Professor Paul and Doctor Wachowski

Guest Editors

Materials

There is a letter below detailing our responses to each of the issues addressed by the reviewers. We hope that the present version of the paper may now be accepted for publication.

Dear Reviewers,

Thank you for your thoughtful comments and helpful suggestions about our manuscript. We have read your comments carefully and detailed modifications have been made accordingly. All the changes are highlighted in yellow in the revised version.

Yours sincerely,

Sebastian Mróz

Reviewer 2 Report

General Evaluation

The paper is an interesting manufacturing study pertaining to SPD of Al1050/MgAZ31/Al1050 sandwich composites by asymmetric ARB process. The investigation is mainly conducted by optical microscopy, mechanical testing, thermomechanical simulation and numerical modeling. The subject is within the scope of the Journal and the manuscript was supported by relevant and recent citations. The presentation of results could be further improved by adding more details concerning the process description and materials characterization, in order to be considered for publication in Materials.

Scientific/Technical Comments

  1. A workflow of the rolling process, including pass schedule and thickness reductions is recommended to demonstrate the major process design and improve the readership and understanding of the process flow. Also, macrographs of the produced composites are recommended to be included.
  2. Total Thickness values and reductions are suggested to be added in Figures 6-11.
  3. The Table 1 seems to be erroneous. The pct % have to be stated as wt. %, while the balances do not add 100% - see 1050 Al.
  4. Figure 1 is of poor quality and not in focus.
  5. The numerical calculations are not comprehensively described in paragraph 2.
  6. The grain size measurements were given in bar chart of Figure 12; however high magnification micrographs of grain structure after rolling process seem that they are missing. It is important to show evidence of progressive grain size refinement and comparison through the microscopic analysis.   
  7. All optical micrographs (at least the lower magnification ones) have to be properly marked showing the different metallic layers.
  8. Some symbols have to be well defined, e.g. the ratio av.

Language/Grammar Comments

The language is in general sufficient. However, there are minor spelling errors that should be corrected during a final proof reading. For instance:

-        L191: Coulomb-Tresca (should be written)

-        In Figure 12, the Y-axis units should be μm.  

Author Response

(The authors gave the same response as above.)

Reviewer 3 Report

The reviewer comments of the paper «Effect of Asymmetric Accumulative Roll-Bonding process on microstructure and strength evolution of an AA1050/AZ31/AA1050 multilayered composite materials»

- Reviewer

The authors presented an article «Effect of Asymmetric Accumulative Roll-Bonding process on microstructure and strength evolution of an AA1050/AZ31/AA1050 multilayered composite materials». Reviewed article is very interesting and write at good scientific level. Presentation method is good and in accordance with generally accepted standards in that area.

However, there are several points in the article that require further explanation.

Comment 1:

Demonstrate in the abstract novelty, practical significance. Give quantitative and qualitative of the proposed method in comparison with existing studies.

Comment 2:

In general, the introduction is well written, but it is useful to review the article: doi:10.1016/j.msea.2019.02.016

It is also useful to cite a paragraph in the introduction and briefly review similar works devoted to: 1050A and AZ31 magnesium alloy. Justify the reason for the choice of these particular materials for research.

Comment 3:

In section 2 it is helpful to give the hardness 1050A and AZ31 magnesium alloy. It is also useful to give more physical and mechanical properties in Table 2.

It is useful to give an experimental stand with a blank and an indication of its main elements. The reader should understand the essence of the research.

Redraw Figure 2 in color.

Give the brand and characteristics of the computer used for the simulation.

Give the accepted parameters of the model: type, dimensions of the finite element mesh, etc. Please describe this in more detail. Justify the choice of software for FEM.

Comment 4:

The conclusions need to more clearly show the novelty of the article and the advantages of the proposed method. What is the difference from previous work in this area? Show practical relevance.

Use the format:

* conclusion 1

* conclusion 2

...

All conclusions must be carefully analyzed by the authors.

Conclusions should reflect the purpose of the article.

Comment 5:

It will be useful to add a section of Nomenclature in which to sign all the physical quantities and abbreviations encountered in the article. There are many physical quantities in the text and such a section will help to find the description of the necessary element.

For example,

ε :              Strain rate (s-1)

AARB      : Asymmetric accumulative roll-bonding

etc.

Comment 6:

English needs polish.

The topic of the article is interesting. But the article should be substantially finalized. Only after major changes can an article be accepted for publication in the «Materials».

Author Response

(The authors gave the same response as above.)

Reviewer 4 Report

In the introduction section, some references are referred using numbers (ex. [1]) while others are referred using the name of the authors and numbers (ex. Line 58: “In papers [24, 25] written by Trojanov et al.”). In my opinion, the authors should use only one method for referring significant previous works in the field, either numbers or numbers plus names.

During the metallographic examination performed to determine the initial structure of materials, is stated that the samples were “cut”. What process was used for cutting the samples and what measures were taken in order to avoid the occurrence of hear affected areas which could influence the structure of the materials?

Lines 110-112: “1050A aluminium samples were cut from the sheet with the thickness of 1 mm, and magnesium alloy samples were designed at the Czestochowa University of Technology (Poland)” – it is quite unclear, aluminium samples were cut, while magnesium alloy samples were just “designed” ?

Line 200: Title of figure 3, “3D computer simulation design of the multilayered Al/Mg/Al band rolling process” – could be replaced by a more illustrative title like “diagram/scheme of computer simulation design of the multilayered Al/Mg/Al band rolling process”. The 3D character of figure 3 is quite unimpressive and it is hard to notice that it really is a 3D representation. Moreover, a simple 2D representation of the process will do the same job.

The "Conclusions" section is very short and ends abruptly. It should summarize the full results of the paper (advantages and disadvantages of the proposed approach), not just some particular findings. Also, further research directions are missing.

The English language should also be improved. While there are no significant errors, some words are improperly chosen (there are much better synonyms for them), and the topic of the phrases is not always the best one.

Just an example: Lines 129-130: “The metallographic examination was performed to determine the initial structure of materials applied for research” – used / utilized for research could be better options than “applied”.

Author Response

(The authors gave the same response as above.)

Round 2

Reviewer 2 Report

Thank you for your revisions which address the review comments sufficiently.

Just some minor checks during proofreading:

  1. In Equation (1) the subscripts of velocities do not match with the equation symbols.
  2. Please check again the Table 1: The mass percentages in Al1050 do not add to 100% (i.e. 99.5+0.45+0.4+0.78+0.25… >>100) !
  3. Also check the spelling of the names: should be Coulomb-Tresca (and not Coulumb-Tresca)

Reviewer 3 Report

The authors did a great job of working out the comments. The article can be published in its current form.